# Blood Fluke Infection (Spirorchidiasis) and Systemic Granulomatous Inflammation: A Case Study of Green Sea Turtles (*Chelonia mydas*) on Jeju Island, South Korea

**DOI:** 10.3390/ani14111711

**Published:** 2024-06-06

**Authors:** Da Sol Park, Won Hee Hong, Jae Hoon Kim, Adams Hei Long Yuen, Sib Sankar Giri, Sung Bin Lee, Won Joon Jung, Young Min Lee, Su Jin Jo, Mae Hyun Hwang, Jae Hong Park, Eun Jae Park, Se Chang Park

**Affiliations:** 1Laboratory of Aquatic Biomedicine, College of Veterinary Medicine, Research Institute for Veterinary Science, Seoul National University, Seoul 08826, Republic of Korea; dabol2@snu.ac.kr (D.S.P.); marinevet@hanwha.com (W.H.H.); yhladams@hotmail.com (A.H.L.Y.); ssgiri@snu.ac.kr (S.S.G.); lsbin1129@snu.ac.kr (S.B.L.); cwj0125@snu.ac.kr (W.J.J.); mushhama@snu.ac.kr (Y.M.L.); ssjjone@snu.ac.kr (S.J.J.); ghkdao@snu.ac.kr (M.H.H.); jaehong139@snu.ac.kr (J.H.P.); eunjae.p@snu.ac.kr (E.J.P.); 2Laboratory of Veterinary Pathology, College of Veterinary Medicine and Veterinary Medical Research Institute, Jeju National University, Jeju 63243, Republic of Korea; kimjhoon@jejunu.ac.kr; 3Radiotherapy and Oncology Centre, Gleneagles Hospital Hong Kong, Wong Chuk Hang, Hong Kong SAR, China

**Keywords:** marine turtle, pathology, diagnosis, parasitology, histopathology, wild animal, herpetology, testudines

## Abstract

**Simple Summary:**

This case report offers a comprehensive analysis of the clinical symptoms, radiological findings, and postmortem examinations of three green sea turtles (*Chelonia mydas*) investigated for spirorchiidiasis. These findings highlight the complexity and severity of the disease, emphasizing the presence of systemic, granulomatous inflammation, and its impact on the overall health of sea turtles. This study advocates for continuous research and conservation efforts to mitigate the impact of spirorchiidiasis on marine turtles.

**Abstract:**

Despite the precarious state of marine turtles as a highly endangered species, our understanding of their diseases remains limited. This case report presents a detailed pathological investigation of spirorchiidiasis, a blood fluke infection that poses a substantial threat to marine turtles. This retrospective study examined three cases of spirorchiid-infected sea turtles, specifically, green sea turtles stranded on Jeju Island, South Korea. Premortem examination of the three spirorchiid-infected green sea turtles demonstrated nonspecific clinical symptoms; blood analysis revealed dehydration, malnutrition, and anemia. Computed tomography scans provided insights into severe pulmonary and extrapulmonary manifestations, including the mass present in the joint region. Post-mortem examinations consistently indicated severe lung lesions and systemic manifestations, with histopathological examination confirming the presence of spirorchiid ova across various organs. Despite the global prevalence of spirorchiidiasis in sea turtles, disease severity varies regionally. This report provides a detailed demonstration of the pathology of spirorchiidiasis in sea turtles from Northeast Asia.

## 1. Introduction

Spirorchiid infections are highly prevalent in turtles worldwide, including in Australia, Hawaii, Florida, Egypt, Italy, Indonesia, India, Japan, and Taiwan [1]. However, manifestations of severe diseases appear to demonstrate spatial disparities [2]. The occurrence of chronic and fatal spirorchiid infections raises concerns about mortality, complicating diagnosis with nonspecific clinical signs such as anorexia, cachexia, sunken eyes, and plastron concavity [3]. The diverse pathological effects associated with spirorchiid fluke infection affect various organs and tissues, primarily the aortae and heart chambers where adult flukes are frequently found, eventually causing systemic granulomatous inflammation [1].

To the best of our knowledge, there are no reports on the pathological details of spirorchiidiasis in sea turtles in Northeast Asia. This retrospective case report was conducted to provide insights into the pre- and postmortem examination of spirorchiid infections in South Korea, particularly in the waters surrounding Jeju Island. We aimed to present the clinical symptoms, computed tomography (CT) results, postmortem gross findings, and histopathology of spirorchiid infections.

## 2. Materials and Methods

### 2.1. Specimen Information

This retrospective case study was conducted to investigate the clinical history and postmortem findings of three stranded juvenile green sea turtles (*Chelonia mydas*) with spirorchiid-infection that were admitted to Aqua Planet, Jeju Island, South Korea (Table 1). Lethargy and respiratory distress were noted in all three turtles upon physical examination. During the treatment period, the turtles received a comprehensive therapeutic regimen involving antibiotics, fluid therapy, vitamin complex injections, anti-inflammatory agents, and forced tube feeding. All three turtles succumbed to their conditions and died during the rehabilitation period.

### 2.2. Premortem Examination

Blood samples were collected for use in hematology and biochemistry. Manual assessments were performed for white blood cells, red blood cell counting, and to determine packed cell volume. Hemoglobin level and blood chemical analyses were carried out using the FujiFilm Dri-chem 4000i (Fujifilm Techno Products Co., Ltd., Takematsu, Japan).

CT scanning was conducted in Case 2 and Case 3 using a 16-row, 32-slice helical CT system from Aquilion Lightning (Aquilion Lightning, Canon Medical Systems, Otawara, Japan). The scan was performed at 120 kV, 250 mA, and 2 mm slice thickness, while the scan field of view (sFOV) was set to 39 cm. CT images were then assessed using open-source Digital Imaging and Communications in Medicine (DICOM) viewing software, Horos version 3.3.6.

### 2.3. Postmortem Examination

Turtles were necropsied within a day of their deaths using standard methods for sea turtle postmortem examination [4]. Tissue samples were collected and fixed with 10% neutral-buffered formalin (OCI Company Ltd., Seoul, Republic of Korea) at a 10:1 fixative-to-tissue ratio before being processed in paraffin wax and sectioned. The tissue slides were then stained with hematoxylin and eosin (H&E), a Gram staining kit (modified Brown and Brenn method; Sigma-Aldrich, St. Louis, MO, USA), and a periodic acid Schiff (PAS) stain kit (Muto Pure Chemicals Co., Ltd., Tokyo, Japan).

## 3. Results

### 3.1. Premortem Examination

#### 3.1.1. Clinical Symptoms and Blood Analysis

All three turtles exhibited uniform clinical symptoms, including algal coverage, buoyancy issues, lethargy, dehydration, and emaciation (Body condition score 2), all of which are indicative of nonspecific distress. Table 2 and Table 3 summarize the hematological and serum chemistry data for each turtle. The observed elevation in blood urea nitrogen aligns with clinically observed dehydration, while the parallel reduction in total protein and cholesterol matches the noted malnutrition. Given the context of dehydration, the manual packed cell volume (mPCV) levels in Cases 2 and 3, namely, 11.5% and 24% (normal range: 17–38%), are considered low and may indicate the presence of severe anemia. The concurrent elevation of aspartate aminotransferase and creatine kinase, observed in all three cases, can be associated with factors like muscle damage and infection causes.

#### 3.1.2. Radiological Findings

Using pulmonary window settings, the CT images of Case 2 demonstrated a ground-glass opacification pattern in the left lung and widespread peribronchovascular consolidations in the right lung, indicating the presence of pulmonary infection (Figure 1A). Both Case 2 and 3 exhibited pleural effusion (Figure 1B and Figure 2A). Case 2 revealed two irregular consolidated masses extending from the pleura, with lengths up to 2.5 cm. These were observed in the lung field (Figure 1C). Case 3 displayed multiple pulmonary nodules (Figure 2A). In Case 3, multiple calcified nodules were observed in the intestinal mucosa and gallbladder (Figure 2B). A well-defined mass (0.7 cm [H] × 1.3 cm [W] × 1.1 cm [L]) enclosed by an intramuscular air pocket was observed beneath the nuchal of the carapace (Figure 2C). A mass (1 cm [H] × 3.2 cm [W] × 3.5 cm [L]) was observed in the left humeroulnar and humeroradial joints (Figure 2D). Using bone window settings, a calcified lesion (1.2 cm [H] × 1.5 cm [W] × 2 cm [L]) was observed beneath the right 5th carapace scutes and adjacent to the right ilium (Figure 2E).

### 3.2. Postmortem Examination

#### 3.2.1. Gross Findings

The necropsy of Case 1 revealed a greenish intracoelomic exudate upon theincision of the coelomic cavity, with abdominal organs and omentum being uniformly covered in yellow nodules (<0.5 mm). The lungs and bronchi showed foam accumulation, and serosa from the lower duodenum to the rectum displayed vasodilation and redness. In Case 2, both lungs had abscesses and possible inflammatory nodules, with bronchial foam observed. The spleen exhibited petechial hemorrhages. Case 3’s necropsy showed multifocal granulomas in lung parenchyma and similar yellow nodules on abdominal organs and lung cross-sections. Inflammatory masses were noted in various joints, including the left elbow, carapace, right femoral, iliac, and sternal joints. No adult flukes were noted during the necropsies.

#### 3.2.2. Histopathological Findings

In Case 1, multifocal granulomas were noted in the pleura, with PAS staining confirming the presence of fungal spores and hyphae in the lungs, alongside yellowish ova indicating spirorchiid infection in the lung parenchyma. The heart showed nonsuppurative endocarditis and vasculitis, while the liver and pancreas revealed granulomas with fungal hyphae and necrosis. Severe inflammation and spirorchiid ova were also present in the spleen and beneath the intestinal mucosa, with urate crystals being present in kidney ducts.

Case 2 revealed similar lung parenchyma granulomas (Figure 3B), with evidence of fungal (Figure 3C), bacterial, and spirorchiid developments. Further, there was spleen inflammation, characterized by the presence of spirorchiid ova around blood vessels (Figure 3A). Additionally, Case 2 showed numerous deposits of dark pigmentation in the liver.

Similarly, Case 3 showed the pathology of lungs with a fungal and bacterial presence, yellowish parasitic ova, and associated inflammatory responses. Notably, Case 3 exhibited severe multifocal granulomatous inflammation, with spirorchiid ova being present in most major organs, including the lungs (Figure 4A), spleen (Figure 4B), liver, heart (Figure 4C), kidneys, and digestive system. Severe multifocal granulomatous inflammation was evident in the carapace and bone tissues (Figure 4D).

The ova observed in our study displayed multiple morphological, similarities to those of the spirorchiid ova reported in earlier research [2]. These eggs, measuring up to 100 × 50 µm, are oval-shaped with a yellowish wall (Figure 3A and Figure 4B,C). The presence of blue, round aggregates, which are considered to be miracidia, along with the surrounding granulomatous lesions is consistent with observations in a previous study [2]. These morphological features confirm that the specimens in our study are spirorchiid ova, supporting our diagnostic conclusions.

## 4. Conclusions

The complex respiratory and multi-organ pathologies seen in fatal spirorchiidiasis cases from Korea highlight the importance of a systemic diagnostic approach for the diagnosis, progression, and prognosis of spirochidiasis. Currently, the diagnosis of spirorchiidiasis relies on accurate postmortem pathogen identification [1]. The alignment between CT findings and histopathology demonstrates the potential benefits of premortem imaging for accurate disease assessment. In addition, given that many fungi are ubiquitous organisms and part of the natural microbiota of the respiratory systems of sea turtles and other reptiles, their role in secondary infections is substantial [7]. The severe granulomatous inflammation observed (especially in the spleen of Case 2), along with spirorchiid infection, suggests that parasitic infection may predispose or exacerbate the growth of opportunistic fungal pathogens, thereby leading to secondary infections. This relationship between the presence of spirorchiid ova and secondary fungal infections underscores the need for a comprehensive understanding of host–pathogen interactions in sea turtles in order to better diagnose, manage, and prevent the progression of such complex infections.

In this study, no adult flukes were found, and DNA could not be obtained from the ova, making species-level identification impossible. Further research, including comprehensive necropsies with traditional parasitological examinations, is needed to accurately assess the infection status of this parasitic group in the domestic sea turtle population. Continuous efforts on an international scale are essential for developing comprehensive strategies for monitoring, managing, and conserving marine turtle populations facing the threat of spirorchiid infection.

## Figures and Tables

**Figure 1 animals-14-01711-f001:**
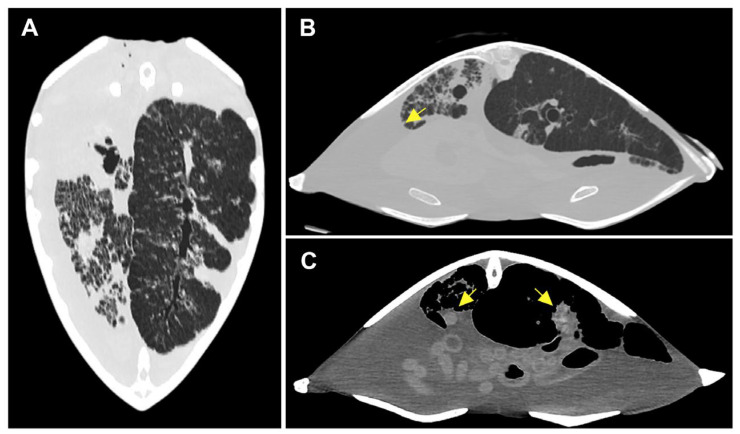
Computed tomography (CT) images of the lung lesion of Case 2. (**A**) Dorsal views showing peribronchovascular and subpleural consolidations were observed in right lung. Opacification of the right lung parenchyma revealed the presence of fluid within the alveolar spaces. (**B**) Axial CT image showing blunting of the costophrenic angle, suggestive of pleural effusion (arrow). (**C**) Axial CT showing a consolidated mass in each lung (arrows). Multiple centrilobular nodules were clearly differentiated within both consolidated masses.

**Figure 2 animals-14-01711-f002:**
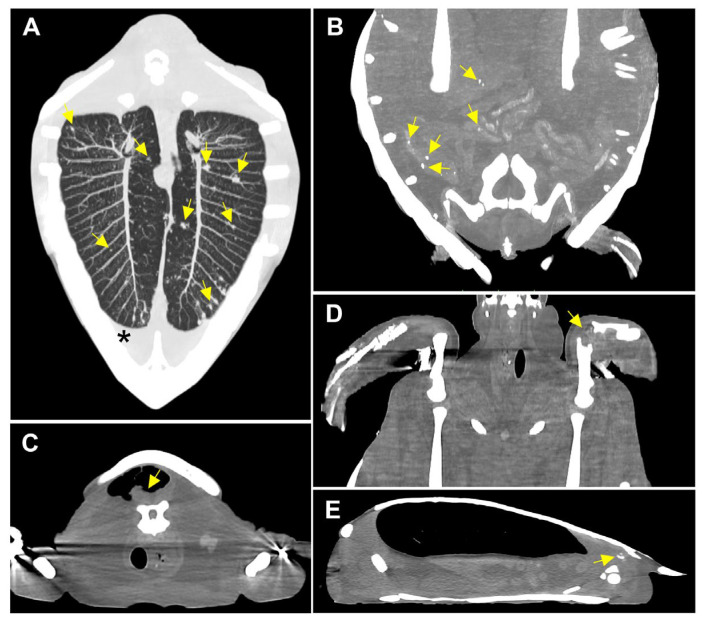
CT images of Case 2. (**A**) Coronal CT image with MIP reconstruction technique revealed the presence of multiple nodules throughout both lungs (arrows). Fluid retention was also noted in both lungs, suggestive of pleural effusion (asterisk). (**B**) Multiple calcified nodules were observed in the intestinal mucosa (arrows). (**C**) CT image showing the presence of a mass just beneath the nuchal of the carapace (arrow). (**D**) CT image showing the presence of a mass at the left humeroulnar and humeroradial joint (arrow). (**E**) CT image showing the presence of a calcified lesion beneath the right 5th carapace scutes and adjacent to the right ilium (arrow).

**Figure 3 animals-14-01711-f003:**
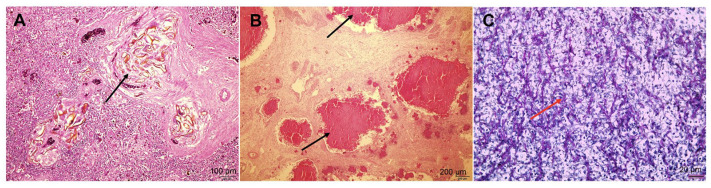
Histopathology of the granulomatous lesions of Case 2. (**A**) Presence of perivascular inflammation and yellowish ova in the spleen (arrow; H&E, ×100). (**B**) Multifocal granulomatous pneumonia in the lung tissue (arrows; H&E, ×40). (**C**) Fungal infection at the center of granulomatous pneumonia lesions (arrow; PAS, ×400). Scale bar = 100 µm (**A**), 200 µm (**B**), 20 µm (**C**).

**Figure 4 animals-14-01711-f004:**
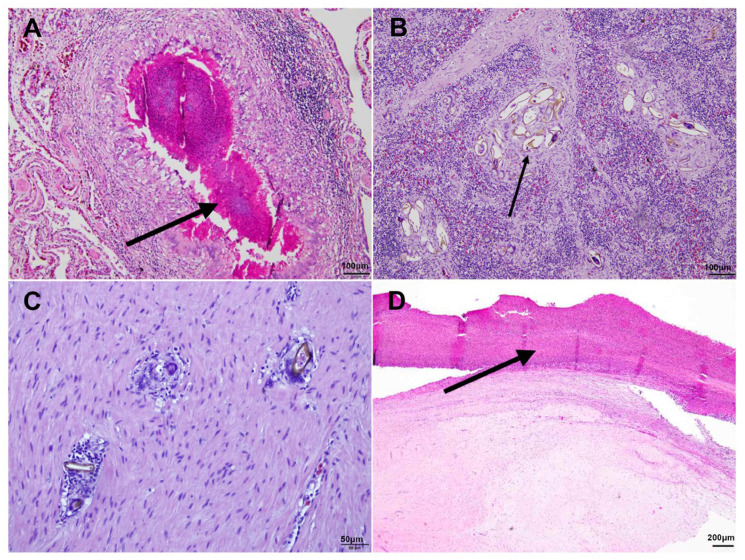
Histopathological findings from Case 3 across multiple organs. (**A**) Multifocal granulomatous pneumonia within the lung tissue (arrow; H&E, ×200). (**B**) Perivascular inflammation and the presence of yellowish parasitic ova in the spleen (arrow; H&E, ×100). (**C**) Parasitic ova embedded within the heart tissue (H&E, ×200). (**D**) Granulomatous inflammation near the cartilage of carapace (arrow; H&E, ×100). Scale bar = 100 µm (**A**,**B**), 50 µm (**C**), 200 µm (**D**).

**Table 1 animals-14-01711-t001:** Detailed case profiles of sea turtles, including age, sex, weight, and a chronological summary of each turtle, covering events from their rescue to the necropsy process.

	Case 1	Case 2	Case 3
Age	Juvenile	Juvenile	Juvenile
Sex	Unknown	Unknown	Unknown
Weight	10.1 kg	9.84 kg	13 kg
Stranded Location	33°26′13.6″ N 126°55′23.4″ E	33°14′19.4″ N 126°36′25.1″ E	33°14′20.9″ N 126°33′43.2″ E
Stranded Date	5 July 2016	21 March 2021	23 May 2022
Date of Death	16 July 2016	26 June 2021	24 June 2022
Date of Necropsy	16 July 2016	26 June 2021	25 June 2022

**Table 2 animals-14-01711-t002:** Hematological profiles of three cases, all assessed before any treatment. The reference range for red blood cell (RBC) values was derived from Samour et al. [5]; other parameters followed the guidelines of Lewbart et al. [6]. The white blood cell (WBC) and RBC levels of Case 1 were unmeasured.

Hematological Parameters	Reference	Unit	Case 1	Case 2	Case 3
WBC	1.76–22.4	10 × 9/L	-	2.8	2.2889
RBC	0.28–0.64	10 × 12/L	-	0.24	0.248
Hemoglobin	5.8–12.9	g/dL	12.3	4.3	9
mPCV	17–38	%	37.5	11.5	24

**Table 3 animals-14-01711-t003:** Blood chemistry profile of all three cases. The reference range for magnesium and creatine kinase values are derived from Samour et al. [5]; other parameters follow the guidelines of Lewbart et al. [6].

Chemistry Parameters	Reference	Unit	Case 1	Case 2	Case 3
Sodium [Na^+^]	157–183	mmol/L	157	143	151
Potassium [K^+^]	4.1–6.9	mmol/L	3.2	2.8	3.8
Chloride [Cl^−^]	100–130	mmol/L	116	95	98
Calcium [Ca^2+^]	1.6–12.2	mg/dL	6.4	1.5	6.5
Phosphorus—inorganic	3.8–10.9	mg/dL	10.4	7.8	10.5
Magnesium [Mg^2+^]	3.8–23.5	mg/dL	7	5.9	7
Blood urea nitrogen	2–37	mg/dL	140	199.6	272.9
Creatinine	0.3–0.9	mg/dL	0.2	0.11	0.15
Uric acid	0.5–3.5	mg/dL	1.2	2.6	3.3
Protein—total	2.6–6.9	g/dL	2.4	1.9	2
Albumin	0.6–2.1	g/dL	0.7	0.6	0.9
Globulin	1.9–5.2	g/dL	1.7	1.3	1.1
Glucose	87–167	mg/dL	88	15	87
Cholesterol—total	73–365	mg/dL	57	14	35
Aspartate aminotransferase	31–389	U/L	358	157	565
Creatine kinase	142–1770	U/L	>2000	68,356	47,384

## Data Availability

Key data supporting this study’s conclusions are included in this publication. Further information and original data are available upon request from the corresponding author.

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
