# Peer review of "Blood Fluke Infection (Spirorchidiasis) and Systemic Granulomatous Inflammation: A Case Study of Green Sea Turtles (Chelonia mydas) on Jeju Island, South Korea"

_animals, 2024, doi:10.3390/ani14111711_

Round 1

Reviewer 1 Report

Comments and Suggestions for Authors

The manuscript described three cases of consequences of infection by spirorchids in stranded juvenile green turtles in South Korea.

The paper is interesting and well organized. However, some important information is lacking: what treatments these alive stranded individuals received in the rearing facility before death?

Minor required change required in the manuscript:

Line 60: Chelonia mydas must be italicized.

Lines 67, 74: use a non-breaking space between value and units.

Table 1, Correct the year of necropsy date for Case 2.

Line 73, add a non-breaking space after Case.

Table 2: Give the abbreviation of WBC, RBC and mPCV. Is it the same than PCV line 95? Why mPCV and not PCV?

Table 3: use exponent characters for +, ++

References 2 and 6, “mydas” must be without cap letter at the beginning.

Author Response

Dear Reviewer 1,

Thank you for your valuable feedback on our manuscript. Please see the attachment for a detailed point-by-point response to each of your comments.

Best regards,

Se Chang Park

Reviewer 2 Report

Comments and Suggestions for Authors

This paper describes results of pathological examinations of stranded sea turtles, with special concerns on spirorchid blood fluke infection. This report is worth to be published, as it is the first report of this kind in Northeast Asia. I would suggest several points to revise as follows. 1) Upon examination of sea turtles, blood fluke infection was expected, as such infection is well known. However, no mention on the examination of adult flukes in the vascular systems in this paper. If the authors examined for the flukes, their presence or absence should be stated. 2) It is quite certain that the yellowish substance was ova of spirorchid flukes, but the authors did not show any ground that they are the ova. The best thing would be molecular identification as shown in Chapman et al. (2019), but it would be enough to compare similar photos of parasite ova shown in references like Chapman et al. (2019). 3) Followings are minor points to be considered.

L60: Chelonia mydas > Chelonia mydas

Table 1: Up to 3 months from the date of stranding to the date of death. How were the turtles kept during that period?

Fig. 1 (A): better to indicate that it is a dorsal view.

L112: Two > two; “two irregular consolidated masses” should be indicated with arrows.

L114: “multiple calcified nodules” should be indicated with arrows. Fig. 2B may be too low in magnification to recognize the nodules.

L118: Figure 2 > Figure 2D; “A mass” should be indicated with an arrow.

L120: Figure 2 > Figure 2E; “a calcified lesion” should be indicated with an arrow.

L131: arrow > arrows

L147: various joints > in various joints

L152-153: alongside yellowish parasitic ova and spirorchiid infection indicators in the lung parenchyma > alongside yellowish ova indicating spirorchiid infection in the lung parenchyma

L157-160: “fungal (Figure 3B), ,,, , plus spleen inflammation characterized by spirorchiid ova around blood vessels (Figure 3C)”: in the legend of Figure 3, 3A and 3C show spleen inflammation and fungal infection, respectively??

L162: inflammatory responses, including spirorchiid ova: inflammatory responses to spirorchiid ova???

L169: parasitic ova > ova

Figure 3: No explanation is given for the arrows in the figures.

L188. 191: secondary diseases > “secondary infections” may be better.

L217: Dis. Aquat. 2019, 133, 217–245.; DOI: 10.3354 > Dis. Aquat. Org. 2019, 133, 217–245.; DOI: https://doi.org/10.3354/dao03348

Author Response

Dear Reviewer 2,

Thank you for your thorough review and constructive feedback. Please see the attachment for a detailed point-by-point response to your comments.

Best regards,

Se Chang Park

Round 2

Reviewer 1 Report

Comments and Suggestions for Authors

The authors answered to the query but they should be more precise about treatment:

"During the treatment period, the turtles received a comprehensive therapeutic regimen in-volving antibiotics, fluid therapy,  vitamin  complex  injections,  anti-inflammatory  agents,  and  forced  tube  feeding. Unfortunately,  all  three  turtles  succumbed  to  their  conditions  and  died  during  the rehabilitation period."

What antibiotics, what dose ? What vitamins and dose ? What anti-inflammatory  agents ?

These precisions are important to establish the protocole to cure this disease: this protocole does not work !

Author Response

Dear Reviewer1,

Thank you very much for your valuable feedback. Please see the attachment for our detailed response.

Best regards,

Se Chang Park
Laboratory of Aquatic Biomedicine
College of Veterinary Medicine and Research Institute for Veterinary Science
Seoul National University
Seoul, Republic of Korea
parksec@snu.ac.kr

Reviewer 2 Report

Comments and Suggestions for Authors

I think corrections have been made properly.

Author Response

Dear Reviewer2,

Thank you very much for your positive feedback. We appreciate your time and effort in reviewing our manuscript. We are glad to hear that you think the corrections have been made properly.

Best regards,

Se Chang Park
Laboratory of Aquatic Biomedicine
College of Veterinary Medicine and Research Institute for Veterinary Science
Seoul National University
Seoul, Republic of Korea
parksec@snu.ac.kr